# Emerging Targeted Therapies in Non-Small-Cell Lung Cancer (NSCLC)

**DOI:** 10.3390/cancers17030353

**Published:** 2025-01-22

**Authors:** Syeda A. Mina, Mohamed Shanshal, Konstantinos Leventakos, Kaushal Parikh

**Affiliations:** 1Division of Hematology and Oncology, Mayo Clinic, Rochester, MN 55905, USA; 2Department of Oncology, Mayo Clinic, Rochester, MN 55905, USA

**Keywords:** targeted therapies 1, NSCLC 2, lung cancer 3, EGFR 4, ALK 5, KRAS 6, MET 7, HER2 8, RET 9, TROP-2 10

## Abstract

Lung cancer remains the leading causes of cancer-related deaths worldwide. The advancement of molecular testing has allowed the identification of oncogenic driver mutations, which has revolutionized the treatment potential for non-small-cell lung cancer (NSCLC). This review article focuses on summarizing the emerging targeted therapies for commonly encountered actionable mutations in NSCLC. We review state-of-the-art treatment strategies for patients with actionable molecular targets in KRAS, EGFR, HER2, ALK, ROS1, MET, RET and FGFR. Additionally, we also discuss novel therapeutic strategies for these targets as well as for TROP2, a frequently expressed cell surface receptor in NSCLC.

## 1. Introduction

Lung cancer continues to be among the most common cancers worldwide and remains the leading cause of cancer-related deaths [1,2]. The broad classification of small-cell lung cancer and non-small-cell lung cancer (NSCLC) remains relevant to date. However, with the advent of advanced testing, including next-generation sequencing (NGS), the landscape of molecular subtypes has evolved, offering several oncogenic drivers as therapeutic possibilities. Over the last two decades, the treatment of lung cancer has undergone a revolution, with immune checkpoint inhibitors and molecular targeted therapies becoming cornerstones of treatment alongside chemotherapy. Mutations in the epidermal growth factor receptor (EGFR) and rearrangements of anaplastic lymphoma kinase (ALK), c-ros oncogene 1 (ROS1), and v-Raf murine sarcoma viral oncogene homolog B1 (BRAF) are the earliest actionable genomic alterations to be rigorously studied, showing significant clinical benefits [3].

From 2003 to 2020, targeted therapies for EGFR, ALK, ROS1, and BRAF advanced substantially, with third-generation EGFR tyrosine kinase inhibitors (TKIs), such as osimertinib, and ALK and ROS1 TKIs, like lorlatinib, now approved for first-line treatment [4,5,6]. Unfortunately, these remarkable discoveries are not pertinent for patients without the respective oncogenic drivers. Over the past decade, substantial effort has been dedicated to discovering novel oncogenic mutations. In the past five years, this has translated into the FDA approval of several newer therapies and ongoing clinical trials.

In this review, we focus on pertinent emerging therapies for known actionable genomic alterations. The molecular biology and established treatments are summarized based on the most current literature and guidelines. Additionally, the latest and maturing data on emerging therapies are primarily collected from interval trial updates and recent conference presentations (ASCO, AACR, WCLC, etc.). Kirsten rat sarcoma viral oncogene homolog (KRAS), EGFR, human epidermal growth factor receptor 2 (HER2), ALK, mesenchymal–epithelial transition (MET), rearranged during transfection (RET), ROS1, and fibroblast growth factor receptor (FGFR) are discussed. Additionally, we also discuss developments in targeting the surface receptor trophoblast cell surface antigen 2 (TROP-2) (Table 1). The purpose of this review paper is to highlight the most active areas of practice-changing innovation within NSCLC. The RAF and MEK pathways are out of scope for this review.

## 2. KRAS

The KRAS protein is a small GTPase encoded on chromosome 12.p12.1, which plays an essential role in intracellular signaling. Active GTP-bound RAS promotes the RAS/RAF/MEK/ERK mitogen-activated protein kinase (MAPK) pathway or PI3K/AKT/mTOR pathway, leading to cell proliferation and survival [7,8]. Mutations in the KRAS gene, particularly in codons 12, 13, and 61, lead to constitutive activation of the KRAS protein, resulting in persistent signaling that drives oncogenesis. KRAS mutations are found in approximately 30% of lung adenocarcinoma cases [9,10,11]. Among these, KRAS G12C is the most common variant, accounting for about 40% of KRAS mutations in lung cancer [12]. Historically, KRAS was considered “undruggable” due to the lack of a suitable binding pocket and its strong affinity and avidity to picomolar concentrations of GTP [13]. The breakthrough discovery of KRAS inhibitors in 2013 was a significant milestone in oncology. The drug is able to covalently and irreversibly bind to the cysteine (Cys) residue of KRAS G12C, which locks the KRAS protein in its inactive or “OFF” GDP-bound state, preventing downstream signaling pathways [13].

CodeBreak 100 was the first study to evaluate sotorasib, a covalent KRAS G12C inhibitor, in advanced solid tumors, including NSCLC. This pivotal, multicenter, single-agent phase I/II trial included locally advanced or metastatic NSCLC after progression on prior therapies. In NSCLC, the overall response rate (ORR) was approximately 37% (95% CI, 28.6–46.2), where the responses were durable, with some lasting beyond 12 months. The median progression-free survival (PFS) was 6.8 months (95% CI, 5.1–8.2), whereas the median overall survival (OS) was 12.5 months (95% CI, 10.0-NE) [14]. Sotorasib was overall well tolerated, with the most common adverse events (AEs) being nausea, fatigue, arthralgia, and an elevated alanine aminotransferase level (ALT) and aspartate aminotransferase level (AST).

Adagrasib is another covalent KRAS G12C inhibitor that was evaluated in NSCLC with similar reported efficacy. The phase II cohort of the KRYSTAL-1 trial for adagrasib demonstrated an ORR of 43% (48/112) among pretreated patients, with a median duration of response (DOR) of 8.5 months (95% CI, 6.2–13.8). The median PFS was 6.5 months (95% CI, 4.7–8.4), and the median OS was 12.6 months (95% CI, 9.2–19.2) [15,16]. In previously untreated brain metastases, the intracranial ORR was 42% (95% CI, 20.3–66.5), making adagrasib the first KRASG12C inhibitor to demonstrate intracranial activity prospectively [17]. The most common AEs observed with adagrasib included nausea, diarrhea, fatigue, musculoskeletal pain, hepatotoxicity, and renal impairment [16].

These findings subsequently led to corresponding pivotal phase III clinical trials. CodeBreaK 200 compared the efficacy and safety of sotorasib to those of docetaxel in previously treated advanced KRAS G12C-mutant NSCLC patients. The ORR was higher in the sotorasib group at 28% compared to 13% in the docetaxel group. Sotorasib showed a statistically significant improvement in PFS compared to docetaxel, with a median PFS of 5.6 months (95% CI, 4.3–7.8) vs. 4.5 months (95% CI, 3.0–5.7), respectively (HR 0.66; 95% CI, 0.51–0.86, *p* = 0.0017). However, there was no significant difference in OS between sotorasib and docetaxel, 10.6 vs. 11.3 months, respectively (HR 1.01; 95% CI, 0.77–1.33) [18]. In the phase III KRYSTAL-12 trial, adagrasib improved PFS compared to docetaxel (5.5 vs. 3.8 months; HR 0.58; 95% CI, 0.45–0.76], *p* < 0.0001). The ORR was higher in the adagrasib group compared to docetaxel at 32% (95% CI, 26.7–37.5) vs. 9% (95% CI, 5.1–15.0), respectively [19].

Several novel KRAS G12C inhibitors are currently in varying stages of clinical trials. Divarasib is one of the novel KRAS G12C inhibitors with greater potency and approximately fifty times more selectivity compared to sotorasib or adagrasib [20]. A phase I study with single-agent divarasib in solid tumors demonstrated a 53% ORR in a subset of NSCLC patients (95% CI, 39.9–66.7) [21]. No new safety concerns were identified, and the most common AEs included nausea, diarrhea, and vomiting [21]. There is an ongoing phase III clinical trial comparing the efficacy of divarasib vs. sotorasib or adagrasib (Krascendo 1, NCT06497556) that will further help clarify its potency and clinical efficacy. Opunarasib is another novel KRAS G12C inhibitor that binds under the switch II pocket (S-IIP) and irreversibly traps KRAS G12C in its OFF state. In preclinical models, it potently inhibited KRAS G12C cellular signaling and proliferation in a mutant-selective manner and demonstrated dose-dependent antitumor activity [22]. A phase Ib/II trial with opunarasib demonstrated an ORR of 42% in the dose escalation cohort of NSCLC patients [23]. Additional clinical trials are in the pipeline with opunarasib. KontRASt-02 is a phase III trial evaluating the efficacy of opunarasib vs. docetaxel in advanced NSCLC [NCT05132075], whereas KontRASt-03 is evaluating opunarasib in select combinations with tislelizumab (PD-1 antibody), trametinib (MEK inhibitor), ribociclib (cyclin-dependent kinase 4/6 inhibitor), or cetuximab (EGFR antibody) [NCT05358249]. KRAS G12C inhibitors such as olomorasib, garsorasib, and others are being assessed and remain in various phases of clinical trials [24,25].

Multiple resistance mechanisms to sotorasib and adagrasib have been implicated, which include the development of adaptive feedback reactivation of RAS signaling pathways such as RAS-MAPK, the activation of receptor tyrosine kinases through MET amplification, the activation of insulin-like growth factor, or the downregulation of E-Cadherin [26,27,28]. Moreover, potentially targetable synthetic lethal genes, including serine–threonine kinases, tRNA-modifying and proteoglycan synthesis enzymes, and YAP/TAZ/TEAD pathway components, have also been identified as mechanisms of resistance [29,30]. On-target and off-target resistance mechanisms cause the plateau of the efficacy of the current KRAS G12C inhibitors. To improve the efficacy of KRAS inhibitors, novel mechanisms of inhibition and KRAS degraders are under investigation.

The discovery and success of KRAS G12C inhibitors revived the search for therapies targeting other KRAS mutations. The most frequently identified KRAS driver mutations in solid tumors include G12D, G12V, G12S, G12R, and Q61H [31]. In NSCLC, after KRAS G12C, the most prevalent KRAS mutations include KRAS G12V and G12D, which constitute approximately 19% and 15% of all KRAS mutations, respectively [32]. Due to the lack of an active Cys residue with these mutations, a more novel mechanism of inhibition was warranted.

In KRAS G12D mutations, the aspartic acid (Asp) residue is significantly less reactive, making it difficult to form covalent bonds. MRTX-1133 is one of the first potent, selective KRAS G12D inhibitors that binds noncovalently to the S-IIP of the active and inactive states of KRAS G12D [33]. Preclinical studies showed significant efficacy, with tumor regression in xenograft mouse models occurring in a dose-dependent manner [34]. These promising preclinical data led to the ongoing phase I/II study of MRTX1133 in KRAS G12D-mutant solid tumors [NCT05737706]. HRS-4642 is another potent, noncovalent KRAS G12D inhibitor with preliminary phase I data available, including 18 patients with approximately three lines of prior treatment. Of them, 33% of the patients had target lesion shrinkage, which included lung adenocarcinoma and colorectal cancer patients [35]. Further investigation on how to enhance the efficacy of HRS-4642 identified proteasomes to be a sensitization target where proteasome inhibitors, like carfilzomib, enhance the antitumor efficacy [36]. These emerging therapies are promising as monotherapies, but their full potential is yet to be determined by identifying the synergistic effects of other therapies. LY3962673 is another KRAS G12D inhibitor that binds noncovalently to the inactive GDP-bound state. In a variety of xenograft models with KRAS G12D-mutant tumors, LY3962673 showed meaningful antitumor activity, making it another promising agent to be used as monotherapy or in combination with other drugs [37]. There are several other noncovalent KRAS G12D inhibitors with promising preclinical data that are currently in different phases of development, including but not limited to drugs such as QTX3046 and INCB161734 [38,39].

The KRAS G12D mutant protein remains in its GTP-bound active (ON) state longer than in its OFF state [40]. Therefore, developing selective RAS-ON inhibitors took precedence but also met significant challenges. The high affinity of the active RAS protein to GTP and shallow binding pockets warranted an innovative molecular design. To overcome these challenges, a tricomplex RAS-ON inhibitor was developed that acts by binding to a chaperone protein, intracellular cyclophilin A, which enables binding to the S-IIP of RAS in a cooperative fashion, allowing covalent linkage with the Asp residue [41,42]. RMC-9805 is a tricomplex KRAS G12D-ON inhibitor that demonstrated significant tumor regression in KRAS G12D-mutant solid tumors, including NSCLC xenograft models, and is now undergoing a phase I trial to better evaluate its safety and efficacy [41].

Another novel approach for targeting mutant KRAS includes KRAS degraders. They are designed as bifunctional molecules, commonly referred to as “PROTACs” (proteolysis targeting chimeras), that have one part that binds to the mutant KRAS protein and another part that recruits a ubiquitin ligase enzyme. The ubiquitination of the KRAS protein then allows it to be recognized by proteasomes for degradation [43,44]. ASP3082 is a novel KRAS G12D protein degrader that has shown preclinical efficacy with dose-dependent antitumor activity in KRAS G12D-mutant NSCLC mouse models [45]. The first-in-human phase I study for ASP3082 is currently ongoing [NCT05382559].

More recently, pan-KRAS inhibitors have been designed to target a broader range of KRAS mutations. These inhibitors preferentially bind to the inactive state of the KRAS protein and work by blocking nucleotide exchange, thereby preventing the activation of both wild-type and mutant KRAS proteins (including G12A/C/D/F/V/S, G13C/D, V14I, L19F, Q22K, D33E, Q61H, K117N, and A146V/T) [46]. Most pan-KRAS inhibitors are currently under investigation in either preclinical or phase I studies. QTX 3034 is one of the noncovalent pan-KRAS inhibitors that bind to the OFF forms of KRAS. In preclinical studies, it was most potent against G12D, with meaningful antitumor activity against G12V as well. It is currently being evaluated in phase I studies for G12D-mutated solid tumors [47,48]. BI-3706674 is another pan-KRAS inhibitor that noncovalently binds to the OFF state of KRAS, with the most notable activity against tumors harboring KRAS G12V mutations and KRAS wild-type amplifications [49]. This is currently under investigation in the phase I setting [NCT06056024]. Moreover, JAB-23425 is another highly potent, orally bioavailable pan-KRAS inhibitor, which can bind to multiple KRAS mutations, including KRAS G12D, G12V, G13D, G12A, G12R, and Q61H, as well as wild-type KRAS in both the ON and OFF states. In a preclinical study, it significantly reduced downstream ERK phosphorylation, inhibiting growth potential in KRAS-mutant cells. It showed high specificity to KRAS while sparing HRAS and NRAS inhibition [50]. The safety profiles of these novel drugs are yet to be determined.

The mechanism of RAS-ON inhibitors that were first identified for the KRAS G12D mutation was later extended to targeting multiple KRAS mutations. For example, RMC-7977 is a reversible, tricomplex RAS-ON inhibitor that has shown broad-spectrum activity against both mutant and wild-type RAS variants. It has demonstrated potent activity in preclinical models of RAS-addicted tumors, particularly those with KRAS codon 12 mutations (KRAS G12X) [51]. RAS-ON inhibitors offer a potential therapeutic option for a wide range of KRAS-mutant solid tumors and those resistant to prior specific KRAS inhibitors.

Furthermore, the discovery of the KRAS G13D pocket has significant implications for the development of both pan-KRAS and RAS-ON inhibitors. The G13D mutation in KRAS is one of the more challenging targets due to its unique structural conformation. Recent studies have identified a novel druggable space in the KRAS G13D mutant protein, characterized by an open P-loop conformation that exposes the catalytic core, making it accessible for targeted inhibition [52]. In the context of pan-KRAS inhibitors, this discovery is crucial, as it expands the potential for these inhibitors to target a broader range of KRAS mutations, other than G13D.

In addition to unlocking the potential of distinct types of KRAS inhibitors, there are simultaneously ongoing clinical trials to evaluate the effectiveness of combination therapies. Strategies for combination therapies involve blocking simultaneous pathways that enable KRAS-mediated oncogenic signaling. Among them, a notable target would be Src-homology region 2-containing protein tyrosine phosphatase 2 (SHP2), which acts as a critical signaling mediator, as it forms a complex with Grb2, Gab1, and SOS1, resulting in the full activation of RAS and its downstream effectors [53]. SHP2 inhibitors, such as SHP099 and hexachlorophene, function by disrupting the formation of the complex, thereby inhibiting downstream signaling pathways [54]. This inhibition leads to reduced cell proliferation, the induction of apoptosis, and the suppression of metastasis in KRAS-mutant NSCLC cells [55]. Several SHP2 inhibitors, including TNO155, RMC-4630, RLY-1971, and JAB-3068, have been under evaluation as single agents as well as in combinations with KRAS and other MAPK pathway inhibitors. Unfortunately, further development of TNO155 and RLY-1971 has been discontinued due to a lack of efficacy and increased toxicity. Early clinical studies have demonstrated that RMC-4630 exhibits clinical activity in patients with KRAS-mutant NSCLC [56]. For instance, a study presented in 2019 reported that RMC-4630 showed promising results in this patient population but highlighted toxicity concerns, such as skin rashes, fatigue, and diarrhea [56]. Moreover, adaptive mechanisms of resistance to combined KRASG12C/SHP2 inhibitors have been identified, including KRAS G12C amplification and alterations in the MAPK/PI3K pathway [30]. Despite these challenges, SHP2 inhibitors remain a promising class of drugs, and perhaps we need to emphasize other avenues of combination therapies. Efforts to refine dosing strategies and patient selection are also critical to overcoming toxicity hurdles and advancing the clinical development of these agents. Ongoing trials include a wide variety of combinations of KRAS inhibitors with immunotherapies, MEK inhibitors, SHP 2 inhibitors, and others, some of which are summarized in Table 2 and Figure 1.

Overall, KRAS inhibitors represent a historical breakthrough in precision oncology. Currently, sotorasib and adagrasib are established standards of care for advanced NSCLC with KRAS G12C mutations for second-line therapy. Data supporting adagrasib are stronger for patients with CNS metastases, and its combinations with immunotherapy are found to be safer compared to sotorasib. Despite their initial success, first-generation KRAS inhibitors have already faced setbacks, such as limited durability in response and the development of resistance mechanisms. As highlighted by the numerous ongoing clinical trials, over the next few years, there will be continued development of different therapeutic strategies to target KRAS, and we predict that a new generation of agents will likely gain preference over existing therapies.

## 3. EGFR

EGFR is a transmembrane receptor tyrosine kinase that plays a critical role in cell growth and survival. Overexpression of EGFR was first recognized in various cancers in the late 1980s and later identified to be an oncogenic driver mutation in NSCLC, along with other cancers [57]. Since EGFR became a promising therapeutic target, EGFR inhibitors have revolutionized the treatment paradigm in lung cancer [57]. First-generation reversible EGFR TKIs, such as gefitinib and erlotinib, were the initial small-molecule inhibitors targeting the ATP-binding site of the EGFR tyrosine kinase domain. The significantly improved PFS and response rates compared to chemotherapy led to their approval. It was notable that certain EGFR mutations (exon 19 deletion (ex19del) and exon 21 L858R point mutation) were particularly responsive to these inhibitors. Nevertheless, despite initial responses, resistance typically developed within a year, often due to the T790M mutation in exon 20 of the EGFR gene, a common gatekeeper mutation [58]. Second-generation irreversible EGFR TKIs, including afatinib and dacomitinib, were designed to overcome resistance by irreversibly binding to EGFR and other members of the epidermal growth factor receptor family. Despite their broader activity, these agents were limited by dose-limiting toxicities and did not significantly improve outcomes in patients with T790M-mediated resistance [59]. Third-generation EGFR TKIs, such as osimertinib, overcame this acquired resistance by specifically targeting the T790M resistance mutation while sparing wild-type EGFR. Subsequently, it became the preferred first-line treatment for EGFR-mutated NSCLC, showing superior outcomes in OS and PFS compared to earlier-generation TKIs [60,61].

More recently, FLAURA-2, an international phase III study, demonstrated the increased efficacy of osimertinib in combination with pemetrexed and platinum-based chemotherapy in advanced NSCLC patients harboring an EGFR E19del or L858R mutation. The primary endpoint of PFS was significantly longer in the combination group at 25.5 months vs. 16.7 months with osimertinib monotherapy (HR 0.62; 95% CI, 0.49–0.79, *p* < 0.001) [62]. The magnitude of the benefit was greater in patients with baseline CNS disease, with a PFS of 24.9 months in the combination group vs. 13.8 months with osimertinib alone (HR 0.47; 95% CI, 0.33–0.66) [63].

There is a continued influx of novel EGFR therapies and combination strategies aimed at targeting resistance mutations and improving patient outcomes. Amivantamab, an EGFR and MET bispecific antibody, was initially approved specifically to target EGFR exon 20 insertion (exon20ins) mutations, which are typically resistant to classical EGFR TKIs [64,65]. Since then, the phase III MARIPOSA and MARIPOSA-2 trials have studied the role of amivantamab and lazertinib, another potent, CNS-penetrant, third-generation EGFR TKI, as either the first line or subsequent line of treatment in advanced EFGR-mutant (ex19del and L858R) NSCLC. In the MARIPOSA study, a combination of amivantamab and lazertinib was compared to either lazertinib or osimertinib monotherapy in the first-line setting. The combination therapy showed a greater median PFS of 23.7 months vs. 16.6 months with osimertinib monotherapy (HR 0.70; 95% CI, 0.58–0.85, *p* < 0.001). Although the ORR was comparable between the two groups, the median DOR was significantly longer at 25.8 months (95% CI, 20.1-NE) with combination therapy vs. 16.8 months (95% CI, b14.8–18.5) with osimertinib. Now, amivantamab–lazertinib is approved as an alternative regimen for the first-line treatment of EGFR-mutant advanced NSCLC. It is important to note that the combination group experienced a higher frequency and grade of AEs, where 75% of patients reported grade 3 or higher AEs. Most notably, they had higher rates of paronychia, rash, infusion-related reactions, and venous thromboembolic events (VTEs) [66]. As the risk of VTEs was greater in the first four months of treatment, the FDA approval recommends four months of prophylactic anticoagulation when initiating treatment with amivantamab plus lazertinib [67,68]. To mitigate the risk of infusion-related reactions, the phase III PALOMA-3 trial studied subcutaneous administration of amivantamab and showed noninferiority compared to intravenous amivantamab in efficacy, with significantly fewer infusion-related reactions (14% vs. 63%) [69].

In the MARIPOSA-2 trial, patients with an ex19del or L858R mutation who progressed on osimertinib were randomized in a 2:2:1 ratio to receive amivantamab–lazertinib–chemotherapy (carboplatin–pemetrexed), amivantamab–chemotherapy, or chemotherapy alone. The response rates were significantly longer in both the amivantamab–lazertinib–chemotherapy and amivantamab–chemotherapy groups, with a longer PFS at 8.3 and 6.3 months (HR 0.44 and 0.48, *p* < 0.001), respectively. In comparison, the median PFS for chemotherapy alone was 4.2 months [70]. Amivantamab-containing groups had a higher frequency of AEs, with the amivantamab–lazertinib–-chemotherapy group having more grade 3 AEs, including neutropenia, thrombocytopenia, rash, and VTEs. Overall, considering the more favorable toxicity profile with improved PFS over chemotherapy, the amivantamab and chemotherapy combination is now the standard of care after the failure of osimertinib.

Similarly, the phase III PAPILLON trial showed the superiority of amivantamab with chemotherapy over chemotherapy alone as the first-line treatment for advanced NSCLC with EGFR exon20ins. The median PFS was 11.4 months for the amivantamab-plus-chemotherapy group, compared to 6.7 months for the chemotherapy-only group, indicating a 40% reduction in the risk of disease progression or death (HR 0.395; 95% CI, 0.30–0.53; *p* < 0.0001) [71].

Given the rapid development of practice-changing data in first- and second-line settings, clinicians need to be more mindful in their interpretation and clinical application of the data. From our perspective, key considerations before initiating therapy include age, functional status, brain metastases, the presence of circulating tumor DNA (ctDNA), co-mutation status, and chemotherapy eligibility [72]. Any presence of brain or liver metastasis, high ctDNA, or co-occurring TP53/RB 1 mutations should warrant a thoughtful discussion regarding combination therapies over osimertinib monotherapy [73]. Notably, a recent press release announced that the combination of amivantamab plus lazertinib improved the median OS by over a year compared to osimertinib. While these data are not available, they are certain to change treatment standards for some patients [74]. Interestingly, the analysis of resistant mechanisms in MARIPOSA showed fewer MET amplifications and secondary EGFR mutations than with osimertinib therapy, which is a result of dual-pathway inhibition [75]. This certainly highlights the importance of resistance testing at the time of progression. Lastly, in older and frail patients with an otherwise limited life expectancy, osimertinib monotherapy still has a significant role. For patients experiencing disease progression on frontline treatment, we advocate for additional tissue and plasma testing to identify potentially actionable resistance mechanisms.

Sunvozertinib is a newer, irreversible EGFR TKI that has high specificity for EGFR mutations, including exon20ins [76]. It was recently granted the FDA breakthrough therapy designation for frontline treatment for patients with EGFR exon20ins-mutated locally advanced or metastatic NSCLC. This was based on the multinational phase II WU-KONG1 trial, which included heavily pretreated advanced NSCLC patients with exon20ins mutations. Sunvozertinib showed a 54% ORR with a disease control rate of 91% with a manageable safety profile with mostly grade 1–2 AEs, including diarrhea, rash, and the elevation of creatine kinase [65]. The ongoing clinical development of sunvozertinib includes confirmatory phase III randomized trials (WU-KONG28) to further establish its efficacy and safety profile [NCT05668988].

Furmonertinib is another third-generation, highly CNS-penetrant EGFR TKI that was first approved in China for the treatment of patients with locally advanced or metastatic NSCLC harboring the EGFR T790M mutation. This was based on the FURLONG study, a phase III trial, where furmonertinib showed superior PFS compared to gefitinib in the first-line setting for EGFR-mutant NSCLC [77]. The phase I study of furmonertinib in EGFR exon20ins advanced NSCLC showed an ORR of 69% and a median PFS of 10 months in treatment-naïve patients in the furmonertinib 240 mg cohort. In comparison, cohorts of previously treated patients receiving furmonertinib 240 mg and 160 mg had a median PFS of 7.0 and 5.8 months, respectively [78]. There is currently a global, phase III study underway for furmonertinib as a first-line treatment in EGFR exon20ins NSCLC patients (FURVENT/FURMO-004; NCT05607550). Currently, the FDA has granted it the breakthrough therapy designation as a potential option for previously untreated EGFR exon20ins-mutated advanced NSCLC.

Despite significant advances in outcomes for EFGR-mutant NSCLC, at the time of progression on EGFR-directed therapy, subsequent lines of salvage therapy remain limited. An alternative targeted therapy is warranted in this setting to overcome EGFR resistance. Human epidermal growth factor receptor 3 (HER3) is overexpressed in many solid tumors and is particularly upregulated in EGFR-mutant NSCLC as a compensatory mechanism leading to resistance to TKIs [79]. This mechanism makes HER3 an attractive therapeutic target. Patritumab deruxtecan (HER3-DXd) is an antibody–drug conjugate (ADC) with a HER3-directed human monoclonal antibody covalently linked to a topoisomerase I inhibitor payload, effectively delivering cytotoxic agents directly to tumor cells overexpressing HER3 [80]. In early-phase studies, patritumab deruxtecan demonstrated encouraging antitumor activity, improving PFS in patients with advanced or metastatic NSCLC [81]. HERTHENA-Lung01 is a phase II trial evaluating HER3-DXd in patients who progressed on prior EGFR TKI and platinum-based chemotherapy. It noted an ORR of 29.8% (95% CI, 23.9–36.2), a PFS of 5.5 months (95% CI, 5.1–5.9), and an OS of 11.9 months (95% CI, 11.2–13.1), with a manageable safety profile and low rate of drug discontinuation due to AEs [82]. Data on further clinical efficacy will be better elucidated through the ongoing HERTHENA-Lung02 phase III trial [NCT05338970]. Novel EGFR-directed therapies, including bispecific and trispecific antibodies, antibody–drug conjugates, and cellular therapies, are under development [83].

Beyond the commonly encountered EGFR mutations discussed above, there is a smaller subset of uncommon EGFR mutations that account for 10–15% of all EGFR mutations [84,85]. Amid the wide range of mutations, G719X, L861Q, S768I, E709X, E709-T710delinsD, and L747X are the most frequently noted uncommon EGFR mutations [86]. Based on structural similarity, these uncommon mutations are classified under one subclass, called P-loop and alpha-helix compressing mutations (PACC), as per the MD Anderson Cancer Center (MDACC) EGFR classification system [87]. These rare mutations have historically shown lower response rates with EGFR TKIs, with moderate sensitivity to second-generation TKIs [86]. A retrospective analysis of subgroups with rare mutations (particularly G719X, S768I, and L861Q) showed significant responses in afatinib-treated cohorts [88]. ACHILLES/TORG1834 is one of the few phase III trials conducted for uncommon EGFR mutations, which also showed significantly longer PFS with afatinib compared to platinum-pemetrexed chemotherapy (10.6 vs. 5.7 months; HR 0.42; 95% CI, 0.26–0.694; *p* = 0.0007) [89]. However, with a lack of CNS activity, there remains an unmet medical need for novel TKIs for EGFR PACC mutant NSCLC. There is an ongoing phase I/II trial investigating ORIC-144, a highly selective, brain-penetrant EGFR/HER2 inhibitor with activity against ex20ins and atypical EGFR mutations, with promising preclinical responses [90].

## 4. HER2

Alterations in HER2, a receptor tyrosine kinase involved in regulating cell growth and differentiation, are present in 1–4% of NSCLC cases [91], and the ERBB2 gene, located on chromosome 17, can be impacted by various genetic abnormalities [91,92]. HER2 alterations in NSCLC include mutations (typically in the kinase domain, exons 18–21), gene amplifications leading to increased HER2 copies, and receptor overexpression identified via immunohistochemistry (IHC), which refers to an excess of HER2 receptors on tumor cell surfaces and plays a key role in selecting targeted therapies, such as trastuzumab deruxtecan (T-DXd), which directly inhibits HER2-driven cancer progression [93].

Initial efforts to target HER2 alterations in NSCLC with pan-HER inhibitors such as afatinib, dacomitinib, and neratinib resulted in disappointing outcomes. ORRs ranged from 0 to 19%, and PFS from 2.8 to 5.5 months [94,95]. The development of novel, selective HER2 inhibitors has shown promise in overcoming the limitations of pan-HER TKIs. Zongertinib is a selective HER2 TKI currently being evaluated in clinical trials. Preliminary results from the Beamion LUNG-1 trial showed a response rate of 50% in patients with solid tumors harboring HER2 mutations, demonstrating the potential of TKIs in this space [96]. An updated analysis of zongertinib in patients with HER2-mutant NSCLC demonstrated an ORR of 72% and a disease control rate (DCR) of 95.5%, with manageable adverse effects, including low rates of severe diarrhea and liver enzyme elevations. Another TKI under investigation is BAY2927088, which selectively targets both HER2 and EGFR mutations. The phase I/II SOHO-01 trial of BAY 2927088 demonstrated promising efficacy in patients with HER2-mutant NSCLC who had progressed after prior systemic therapy. Among 34 treated patients, the ORR was 70%, with responses being rapid and durable. The median PFS was 8.1 months. Additionally, 95% of patients with baseline HER2 ctDNA had a decrease in ctDNA during treatment, indicating significant antitumor activity. The safety profile was manageable, with diarrhea and rashes being the most common AEs [97].

The most significant advancement in the treatment of HER2-altered NSCLC has been the development of HER2-directed ADCs. T-DXd has emerged as the leading therapeutic option, offering unprecedented efficacy in pretreated patients. In the pivotal DESTINY-Lung01 trial, T-DXd achieved a 55% ORR, a median PFS of 8.2 months, and a median OS of 17.8 months [98]. However, interstitial lung disease (ILD), occurring in approximately 26% of patients, remains a concern and necessitates vigilant monitoring. Looking ahead, the DESTINY-Lung04 trial [NCT05048797] aims to compare T-DXd with standard chemotherapy plus immunotherapy in the first-line setting [99]. This trial, along with ongoing research into TKI-based therapies, will help define the role of HER2-targeted treatments in NSCLC. An ongoing trial of SHR-A1811, a novel HER2-targeted ADC, has shown promise in HER2-mutant NSCLC. In the phase 1 study, SHR-A1811 demonstrated an ORR of 41.9% and a median PFS of 8.4 months in patients with HER2-mutant NSCLC [100].

Monoclonal antibody-based therapies have also been explored in HER2-mutant NSCLC. The combination of trastuzumab, pertuzumab, and docetaxel has shown encouraging results, with an ORR of 29% and a median duration of response of 11 months [101]. While these results warrant further investigation, monoclonal antibodies have not yet matched the efficacy seen with ADCs like T-DXd.

HER2 bispecific T-cell engagers (BiTEs) engage HER2-expressing tumor cells with T cells to promote immune-mediated killing. Preliminary studies suggest that HER2 BiTEs could be effective in NSCLC, although further research is needed to confirm their safety and efficacy in larger patient cohorts. GBR 1302, a HER2xCD3 bispecific antibody, is being evaluated in an ongoing phase 1 trial (NCT02829372) for patients with HER2-positive solid tumors. Among the 19 evaluable patients, infusion-related reactions (IRRs) and cytokine release syndrome (CRS) were the most common AEs at doses ≥ 100 ng/kg, with one patient experiencing a grade 4 IRR/CRS at 500 ng/kg requiring ICU care but resolving within 36 h. No radiological responses were noted, but two patients (one with HER2 3+ gastroesophageal adenocarcinoma and one with HER2 2+ breast adenocarcinoma) demonstrated prolonged disease stabilization lasting ≥4 months [102]. As HER-2 therapies have a stronger presence in clinical practice, providers must incorporate HER-2 mutation testing as well as expression analysis at the time of diagnosis and progression.

Despite advancements in HER2-targeted therapies for NSCLC, key challenges remain. Toxicity management is a critical issue, particularly the risk of interstitial lung disease (ILD) associated with ADCs like T-DXd. Effective strategies, including dose adjustments and early detection, are essential to reduce these risks and improve patient outcomes. Future directions will focus on enhancing drug design, developing combination therapies, and optimizing treatment sequencing to maximize efficacy and minimize toxicity, particularly in patients with CNS involvement.

## 5. MET

The c-MET (mesenchymal–epithelial transition) proto-oncogene on chromosome 7q21-q31 encodes the receptor protein tyrosine kinase MET, which is responsible for the activation of the MAPK pathway regulating cell proliferation [103]. MET alterations constitute approximately 7% of the targetable mutations within NSCLC [104]. Common MET-driven alterations include MET exon 14 (METex14) skipping mutations, MET amplification, MET overexpression, and MET fusions. The METex14 mutation during pre-mRNA splicing is the most common point mutation. It is responsible for the loss of the tyrosine residue Y1003, which results in the decreased ubiquitination and prolonged half-life of the MET receptor [105,106]. The METex14 mutation is found in 3–4% of lung adenocarcinomas and 1–2% of other lung cancer histologies, except for sarcomatoid histology, where approximately 20% of cases harbor the mutation [107,108]. These mutations tend to be mutually exclusive with other common driver mutations such as EGFR, KRAS, and ALK [109]. Moreover, MET amplifications are characterized by an increase in the number of copies of the MET gene within a cell. This overexpression can activate downstream signaling pathways like the PI3K/AKT/mTOR and RAS/RAF/MEK/ERK pathways [106]. High amplifications are typically defined as having more than 10 copies of the MET gene or a ≥4 MET-to-CEP7 ratio. MET amplifications occur in a smaller subset of NSCLC, approximately 0.78% in adenocarcinomas and 1.07% in squamous cell carcinomas [110]. MET amplification can be a resistance mechanism responsible for 5–20% of EGFR TKI resistance or occur independently of other oncogenic drivers, which usually present as high-level MET amplification [111].

MET inhibition can be achieved through nonselective multikinase inhibitors (MKIs), such as crizotinib and cabozantinib, or selective MET inhibitors, such as capmatinib and tepotinib. The GEOMETRY mono-1 trial led to the approval of capmatinib for the treatment of METex14-mutated metastatic NSCLC. A significant response was noted, with an ORR of 68% (95% CI, 48–84) and a DOR of 12.6 months (95% CI, 5.6-NE) in treatment-naïve patients and an ORR of 41% (95% CI, 29–53) with a DOR of 9.7 months (95% CI, 5.6–13.0) in previously treated patients [112]. The efficacy of MET amplification was more pronounced in tumors with high gene copy numbers. For MET amplification with a gene copy number of >10, the ORR was 40% (95% CI, 16–68) in treatment-naïve patients and 29% in previously treated patients (95% CI, 19–41) [112].

Subsequently, tepotinib was FDA-approved for the treatment of metastatic NSCLC harboring METex14 mutations based on the VISION trial. In treatment-naïve patients, the ORR was 57.3% (95% CI, 49.4–65.0) with a median DOR of 46.4 months (95% CI, 13.8-NE). For previously treated patients, the ORR was 45.0% (95% CI, 36.8–53.3) with a median DOR of 12.6 months (95% CI, 9.5–18.5) [113]. Moreover, in the CHRYSALIS study, amivantamab showed an overall ORR of 33% in patients with primary METex14 whose disease progressed on or who declined the current standard-of-care therapy [114].

Savolitinib is another MET TKI that has shown efficacy in both first- and second-line settings, including patients with aggressive pulmonary sarcomatoid carcinoma. It is currently being evaluated in various clinical trials to further establish its role in treating METex14-mutated NSCLC [107]. The role of other nonselective inhibitors, including cabozantinib, merestinib, and glesatinib, are being assessed in ongoing phase II clinical trials as well (NCT01639508, NCT05613413, NCT04310007, NCT03911193).

Promising new mechanisms to target MET include ADCs such as REGN5093-M114 and Telisotuzumab vedotin (Teliso-V). REGN5093-M114 showed significant antitumor efficacy in preclinical models, including those with METex14mutations, by overcoming MET-driven acquired resistance to EGFR TKIs [115]. A phase I/II study is still underway for MET-overexpressing advanced cancer [NCT04982224]. Teliso-V is another MET-targeted ADC that carried a cytotoxic monomethyl auristatin E payload and was noted to be particularly efficacious in phase I studies for MET-overexpressing NSCLC patients. More recently, the phase II LUMINOSITY study with Teliso-V demonstrated an ORR of 28.6% (95% CI, 21.7–36.2) in patients with previously treated MET-overexpressing advanced nonsquamous EGFR-wild-type (WT) NSCLC. MET overexpression was characterized using immunohistochemistry as >25% tumor cells with 3+ stain, which was further subdivided into high and intermediate. The ORR, DOR, PFS, and OS were proportional to the degree of MET expression [116]. Another novel topoisomerase 1 inhibitor payload conjugated to telisotuzumab, ABBV-400, is also in the pipeline with tolerable safety and efficacy in the phase I setting [117]. As therapies for MET alterations beyond METex14 continue to expand, clinicians need to recognize MET overexpression/amplification for later lines of MET-directed treatment.

## 6. ALK

While anaplastic lymphoma kinase (ALK) gene fusion was seen in large-cell lymphomas, its fusion with the promoter gene, echinoderm microtubule-associated protein-like 4 (EML4), was first discovered in 2007 in NSCLC as an oncogenic driver alteration [118,119]. Since then, ALK fusions have been studied extensively and are seen in 4–7% of all NSCLC [120]. While ALK, as a cell surface receptor tyrosine kinase, is responsible for embryonal nervous system development, it has no clear role later in life and is suppressed in normal cells. This makes ALK an attractive target when detected, as it is expected to be present only in cancer cells. When rearranged with a promoter gene, the fusion protein presents as a cytoplasmic protein that leads to constitutive downstream activation of the MAP-kinase, PI3K-AKT-mTOR, and JAK-STAT pathways, leading to uncontrolled cell cycle progression and carcinogenesis. Crizotinib was the first ALK inhibitor approved for clinical use, but the major drawbacks of this agent were a lack of intracranial activity and the development of ALK resistance alterations [121].

Until recently, second-generation ALK inhibitors such as alectinib, brigatinib, and ceritinib were the preferred frontline agents for ALK-fusion-positive advanced NSCLC [122,123,124]. Lorlatinib, a third-generation ALK TKI, was available for second-line therapy for patients who developed ALK resistance mutations following exposure to second-generation ALK inhibitors, most commonly G1202R [125]. However, results from the phase III CROWN trial led to the approval of lorlatinib as frontline therapy for patients with advanced ALK-positive NSCLC. In this study, 296 patients with metastatic ALK-fusion-positive NSCLC were randomized to receive lorlatinib (n = 149) or crizotinib (n = 147), with PFS as the primary endpoint. The study met its primary endpoint, with median PFS not reached for lorlatinib versus 9.3 months (95% CI, 7.6–11.1) for crizotinib (HR 0.28, 95% CI, 0.19–0.41; *p* < 0.001) [126]. Furthermore, the 5-year PFS rate was 60% with lorlatinib versus 8% with crizotinib [127]. High-risk features such as EML4-ALK variant 3, concurrent TP53 mutations, intracranial metastases, and a high ctDNA concentration are associated with reduced benefits with second-generation ALK TKIs [128]. Contrastingly, in the CROWN trial, the benefit of lorlatinib was sustained in patients with EML4-ALK variant 3 or with concurrent TP53 mutations [127]. Based on these results, lorlatinib is a valid frontline treatment option for patients with ALK-fusion NSCLC. While lorlatinib is currently the most potent ALK inhibitor available, it has a unique adverse effect profile that warrants discussion. Some of the common adverse events seen with lorlatinib are hyperlipidemia, including hypercholesterolemia (all grades: 72%, grade ≥ 3: 21%) and hypertriglyceridemia (all grades: 66%, grade ≥ 3: 25%), edema (all grades: 57%, grade ≥ 3: 4%), weight gain (all grades: 44%, grade ≥ 3: 4%), and cognitive and mood effects (all grades: 49%, grade ≥ 3: 4%). Although ALK inhibitors have shown superior efficacy and better safety profiles compared to either chemotherapy or crizotinib, there is no available head-to-head comparison among them. The current literature best offers data from a systematic review and network meta-analysis ranking the newer ALK inhibitors based on available phase III RCTs. Lorlatinib demonstrated the highest efficacy in terms of PFS and ORR, while low-dose alectinib demonstrated the best safety profile and was most effective for patients with baseline brain metastasis [129]. This further justifies lorlatinib to be the first-line agent in clinical practice now.

The inhibition of tropomyosin receptor kinases A, B, and C (TRKA, TRKB, TRKC) by lorlatinib has been implicated in the cognitive and behavioral changes as well as weight gain associated with lorlatinib [130]. To address the adverse effects associated with lorlatinib, newer ALK TKIs in development, such as deulorlatinib and NVL-655, have shown promise. Deulorlatinib is a third-generation ALK TKI that has a selective deuterium switch at the N-methyl group in the pyrazole of lorlatinib, leading to greater stability and the inhibition of several ALK and ROS variants. In a phase Ia/Ib trial, deulorlatinib showed an ORR of 87.9% in TKI-naïve patients (n = 33) and 38.1% in patients treated with a second-generation TKI (n = 97), with strong intracranial activity [131]. NVL-655 is a novel fourth-generation ALK TKI that was designed to avoid TRK inhibition while maintaining activity against single and compound ALK mutations and intracranial penetrance [132]. Furthermore, it has also shown activity against the compound G1202R/L1196M mutation, which confers resistance to lorlatinib. In a phase I/II study of heavily pretreated patients, NVL-655 showed an ORR of 37% in patients with three prior ALK TKIs, including lorlatinib (16/43), and 53% in lorlatinib-naïve ALK TKI-treated patients (16/43) [133].

## 7. ROS1

The ROS1 gene encodes the receptor tyrosine kinase that is normally found in embryonic tissue, with minimal expression in adulthood [134]. Its oncogenic potential arises from chromosomal rearrangements, with resultant ROS1 fusion proteins that lead to constitutive receptor activation and downstream MAPK, PI3K, and JAK/STAT signaling [135]. ROS1 rearrangements occur in approximately 1–2% of NSCLC, typically in younger patients who are light or never smokers and have adenocarcinoma histology [135,136].

The clinical significance of ROS1 in NSCLC is underscored by the effectiveness of targeted therapies. Crizotinib was the first FDA-approved drug for ROS1-positive NSCLC, demonstrating significant clinical activity with high response rates and prolonged PFS, irrespective of the ROS1 fusion partner [137]. However, most patients eventually relapsed with CNS metastases due to a lack of CNS penetration and the development of resistance mutations [138,139,140]. The newer-generation TKIs, such as ceritinib, lorlatinib, and entrectinib, that have activity against ROS1 have shown better efficacy, along with CNS penetration, and are currently used in clinical practice [141].

Entrectinib was approved based on the integrated analysis of three clinical trials: ALKA-372-001, STARTRK-1, and STARTRK-2 [142]. These trials demonstrated an ORR of 77% (95% CI, 64–88) and a DOR of 24.6 months (95% CI, 11.4–34.8) with entrectinib. It had a relatively manageable safety profile with 34% grade ≥ 3 AEs, the most common being weight gain and neutropenia [142]. More recently, the next-generation ROS1/TRK/ALK inhibitor, repotrectinib, received approval based on the TRIDENT-1 phase II trial. The study demonstrated an ORR of 79% (95% CI, 68–88) and a PFS of 34.1 months (95% CI, 25.6-NE) in TKI-naïve patients versus an ORR of 38% (95% CI, 25–52) and a PFS of 9.0 months (95% CI, 6.8–19.6) in previously treated patients [143]. Repotrectinib is noted to have high potency against ROS1 mutations, including solvent-front mutations like G2032R [144].

Taletrectinib is another next-generation, CNS-penetrant ROS1/TRK inhibitor with a recent FDA breakthrough therapy designation for locally advanced or metastatic ROS1-mutated NSCLC [145]. Data pooled from two pivotal studies, TRUST-I (NCT04395677) and TRUST-II (NCT04919811), highlighted its robust activity in both treatment-naïve and previously TKI-treated patients. The study included 217 patients, where TKI-naïve patients demonstrated an ORR of 92% [95% CI, 85.4–95.7], with PFS not reached. Patients previously treated with TKI had an ORR of 54% [95% CI, 43.0–64.8] with a PFS of 9.6 months (95% CI, 5.7–12.0). The most common AEs were liver enzyme abnormality, dizziness, and diarrhea, with drug discontinuation in 6% of patients [146].

Given the above evidence, entrectinib and repotrectinib are our preferred frontline agents, especially in cases with intracranial metastasis. Emerging combination therapies with ROS1 inhibitors for the treatment of NSCLC include strategies to overcome resistance mechanisms and enhance efficacy. As EGFR and MET pathways are common pathways of resistance [147,148], one notable combination involves amivantamab with TKIs in advanced NSCLC patients harboring ALK, ROS1, and RET gene fusions. This combination is currently under investigation in a phase 1/2 study [NCT05845671].

## 8. RET

The RET (rearranged during transfection) oncogene is located on chromosome 10q11.2, which encodes a receptor tyrosine kinase responsible for cell growth, differentiation, and survival. RET mutations are associated with multiple endocrine neoplasia type 2 (MEN 2) and thyroid malignancies [149]. In 2011, RET rearrangements were identified within NSCLC, suggesting its potential to be an oncogenic driver [150]. Different fusion variants were discovered, of which kinesin family member 5B gene (KIF5B)-RET fusion was the most common [151]. These aberrant RET fusion proteins constitutively activate the RET kinase domain, which in turn activates downstream signaling pathways (e.g., MAPK, PI3K/AKT) that promote cell proliferation and survival [152]. RET fusions are found in 1–2% of NSCLC cases, particularly in adenocarcinomas and in patients who are younger, are non-smokers, or have light smoking exposure [153,154].

MKIs such as cabozantinib, vandetanib, and lenvatinib, which are known to be effective in other malignancies, were tested in NSCLC. These agents demonstrated a limited OS benefit, with substantial off-target side effects due to their nonselective nature, and were primarily used as off-label agents in RET-fusion-positive NSCLC [155,156,157,158].

However, the landscape has remarkably improved with the advent of highly selective RET inhibitors. Selpercatinib is a small molecule that binds to the ATP-binding site of the RET kinase, inhibiting the phosphorylation of RET and its downstream cell signaling. Moreover, it has been shown to cross the blood–brain barrier, providing intracranial efficacy [159]. In the multicenter LIBRETTO-001 trial, selpercatinib activity was assessed in both treatment-naïve and previously treated patients with advanced RET-fusion-positive NSCLC. An ORR of 64% (95% CI; 54–73) was observed, with a median DOR of 17.5 months (95% CI, 12.0-NE) according to the independent review committee and 20.3 months (95% CI, 15.6–24.0) according to the investigator assessment. Given the effective CNS penetrance of selpercatinib, an objective intracranial response of 91% was observed (10 of 11 patients; 95% CI, 59–100). The grade ≥ 3 AEs were primarily hypertension, increased transaminases, hyponatremia, and lymphopenia, with only a 2% rate of drug discontinuation [160]. The phase III LIBRETTO-431 trial comparing selpercatinib to the standard-of-care chemo/immunotherapy regimens demonstrated a median PFS of 24.8 months compared to 11.2 months (95% CI, 0.31–0.70; *p* < 0.001). The ORR was 84% (95% CI, 76–90) in the selpercatinib group, compared to 65% (95% CI, 54–75) in the control group [161].

Similarly, pralsetinib is another small molecule that strongly and selectively inhibits the RET kinase. The ARROW trial was a multicenter, phase I/II trial assessing the efficacy of pralsetinib in multiple cohorts of patients with RET-mutated malignancies. Among cases of RET-fusion-positive NSCLC, there was an ORR of 61% (95% CI, 50–71) in treatment-naïve patients and 70% (95% CI, 50–86) in previously treated patients [162]. This data also led to the accelerated FDA approval of pralsetinib for RET-fusion-positive NSCLC in 2020. The updated safety and efficacy data share a similar ORR of 72% [95% CI, 60–82] and a PFS of 13.0 months for treatment-naïve patients and an ORR of 59% (95% CI, 50–67) with a PFS of 16.5 months for previously treated patients [163]. The drug continues to demonstrate a favorable safety profile, with the most common AEs including neutropenia (18%), hypertension (10%), increased creatine phosphokinase (9%), and lymphopenia (9%) [163]. The ongoing AcceleRET Lung study is an international, open-label, randomized phase III trial (NCT04222972) designed to evaluate the efficacy and safety of pralsetinib compared to the standard of care for the first-line treatment of patients with advanced or metastatic RET-fusion-positive NSCLC [164].

The mechanisms of resistance to pralsetinib and selpercatinib include both RET-dependent and RET-independent pathways, like MET or KRAS amplifications. Among the RET-dependent mechanisms, the best-characterized mutation is the G810X mutation. These are located at the kinase solvent-front site that alters the conformation of the RET kinase domain, leading to steric hindrance and preventing the effective binding of the inhibitors [165,166].

To overcome these resistance mechanisms, several next-generation RET TKIs have been under investigation. TPX-0046, a next-generation RET/SRC inhibitor that had initially shown preclinical efficacy in the SWORD-1 phase I/II study, was eventually terminated due to adverse effects [167]. Another promising agent is ALKynyl nicotinamide-based inhibitors, such as HSN608, HSL476, and HSL468. These compounds have demonstrated potent activity against a range of RET mutations, including the G810 solvent-front mutants and the V804M gatekeeper mutant, in preclinical studies [168].

Lastly, although the incidence of RET alterations is generally low in NSCLC, it is important to consider it for young and nonsmoker patients who may not have any other targetable alterations. Additionally, it is important to recognize the importance of RNA-based testing for RET fusions, as it has been shown to have higher sensitivity and specificity in detecting RET fusions compared to DNA sequencing alone. This is particularly important for identifying noncanonical RET fusions and ensuring accurate diagnosis and treatment planning [169].

## 9. FGFR

The fibroblast growth factor receptor (FGFR) pathway is a tyrosine kinase signaling pathway that promotes cell proliferation, survival, and angiogenesis [170]. FGFR aberrations have also been identified as key compensatory bypass mechanisms of resistance to targeted therapy against EGFR TKIs and KRAS inhibitors [171,172]. Oncogenesis is driven by various mechanisms of FGFR alterations, including amplifications, mutations, and translocations [173]. The FGFR1-3 fusion genes are a unique molecular subset more frequently identified in NSCLC, especially in patients with a prior smoking history [174]. FGFR mutations are more frequently observed in squamous cell lung cancer (6.8%) compared to other NSCLC subtypes (1.3%) [175,176,177].

Currently, no agents targeting FGFR in NSCLC have been approved. However, several FGFR inhibitors have been approved for other cancers, such as pemigatinib and futibatinib, and advanced cholangiocarcinoma with FGFR2 fusions or rearrangements [178]. Erdafitinib has been approved for the treatment of metastatic urothelial carcinoma with FGFR2 and FGFR3 genetic alterations [179].

These agents have shown potential with off-label use in FGFR-mutated NSCLC patients. For example, a patient with stage IV squamous cell lung cancer with FGFR3-TACC3 fusion achieved 11 months of disease control with a radiographic response on erdafitinib [180]. Such clinical experiences highlight the potential of FGFR-targeted therapies within NSCLC. The ongoing open-label, multicenter, phase I study LOXO-435 with FGFR3-altered advanced solid tumors will shed more light on the clinical efficacy in NSCLC soon [NCT05614739].

## 10. TROP-2

Trophoblast cell surface antigen 2 (TROP-2) is a transmembrane protein belonging to the epithelial cell adhesion molecule (EpCAM) family [181]. It was initially identified as a marker of trophoblast cells and later recognized for its significant overexpression in various epithelial cancers while having relatively low or no expression in most normal tissues [181,182]. In 2008, TROP-2 was identified as an oncogene contributing to tumorigenesis, where its inhibition reduced tumor invasiveness, granting it therapeutic potential [183]. Among NSCLC subtypes, it was observed to be prevalent in 75% of squamous cell carcinomas, 65% of lung adenocarcinomas, and 18% of high-grade neuroendocrine tumors [184]. Lung adenocarcinomas with TROP-2 expression had higher lung-cancer-specific mortality [184]. TROP-2 expression allows the activation of several intracellular signaling pathways, including PTEN/PIK3CA/Akt, MAPK/ERK, JAK/STAT, ErbB, TGFβ, and WNT/βcatenin, contributing to cell survival and growth [185]. Therefore, TROP-2 has emerged as a promising therapeutic target due to its high expression in tumors and association with poor prognosis [186]. Therapeutic strategies targeting TROP-2 include ADCs such as sacituzumab govitecan (SG), which has shown clinical efficacy in treating metastatic triple-negative breast cancer and other refractory cancers [187].

SG is an anti-TROP-2 humanized monoclonal antibody conjugated with the topoisomerase inhibitor SN-38, an active metabolite of irinotecan via a cleavable CL2A linker. It has a high drug–antibody ratio (DAR) of 7.6:1, offering greater antitumoral efficacy even with a low antigen density [188]. The initial phase I trial included advanced solid tumors not preselected for TROP-2 expression and showed promising therapeutic activity with infrequent grade ≥ 3 AEs [189,190]. It became the first anti-TROP-2 ADC to be approved for metastatic triple-negative breast cancer based on the single-arm clinical trial results of IMMU-132-01, with an ORR of 33.3% and a median DOR of 77.7 months [191]. A subset of 54 pretreated NSCLC patients was part of a single-arm expansion cohort, where the ORR was 19% (n = 19/47) and 17% in the intention-to-treat population (n = 9/54), with a median DOR of 6.0 months (95% CI, 4.8–8.3) and a median PFS of 5.2 months (95% CI, 3.2–7.1) [192].

In May 2024, the initial results from phase III EVOKE-01 were presented, comparing docetaxel (n = 304) vs. SG (n = 299) in pretreated metastatic NSCLC patients who progressed on platinum-based chemotherapy, immunotherapy, and targeted therapy [193]. Although the primary endpoint of OS was not met, there was a numerical improvement in OS for SG compared to docetaxel (median OS of 11.1 months vs. 9.8 months; 95% CI, 0.68–1.04). The PFS was similar between the two groups (4.1 months for SG vs. 3.9 months for docetaxel, HR of 0.92; 95% CI, 0.77–1.11). Notably, a clinically meaningful OS benefit was observed with SG over docetaxel in patients nonresponsive to their last immunotherapy-containing regimen (median OS of 11.8 vs. 8.3 months, HR of 0.75; 95% CI, 0.58–0.97) [193].

Several ongoing clinical trials are investigating the most effective way to utilize SG in the treatment of advanced NSCLC. EVOKE-02 [NCT05186974] is an ongoing, multicenter, phase II trial evaluating SG with pembrolizumab ± platinum chemotherapy for treatment-naïve metastatic NSCLC patients who have PD-L1 ≥ 50% and no actionable mutations. Early safety data of cohort A (SG+ pembrolizumab) with a median follow-up of 11.3 months reported common grade ≥ 3 TRAEs to be neutropenia, diarrhea, and respiratory failure [194]. The results of ORR and PFS are still to be determined. The EVOKE-03 study is simultaneously ongoing to assess the efficacy of SG with pembrolizumab vs. monotherapy with pembrolizumab for metastatic NSCLC [NCT05609968]. Although the results for SG have not been promising in the pretreated population thus far, we will have to await trials with SG in a first-line setting to better understand the clinical implications.

Another emerging anti-TROP-2 ADC is datopotamab deruxtecan (Dato-DXd), which had promising clinical activity in the phase I TROPION-PanTumor01 study including heavily pretreated patients [195]. This inspired subsequent phase I/II trials evaluating the safety and efficacy of Dato-DXd in patients with advanced or metastatic NSCLC [196].

The global phase III, open-label TROPION-Lung01 study evaluated Dato-DXd vs. docetaxel in patients with previously treated, advanced NSCLC [197]. The median PFS difference was statistically significant between Dato-DXd (n = 299) and docetaxel (n = 305) at 4.4 vs. 3.7 months, respectively (HR of 0.75; 95% CI, 0.62- 0.91, *p* = 0.004). There was a greater difference noted with nonsquamous histology, with a median PFS of 5.5 vs. 3.6 months, respectively (HR of 0.84; 95% CI 051–0.79). However, there was no difference in OS between Dato-DXd and docetaxel (7.6 vs. 9.4 months, HR of 1.32; 95% CI, 0.91–1.92) [197]. Given these disappointing results, the biologic licensing application for Dato-DXd was withdrawn for second-line therapy. The negative results were surprising, instigating further analysis to identify better predictive biomarkers than TROP-2 expression. In this vein, Garassino et al. reported the association between the cell membrane and cytoplasmic TROP-2, expressed as a normalized membrane ratio (NMR) measured by quantitative continuous scoring (QCS), and outcomes in the TROPION-Lung01 study. Interestingly, they identified that TROP-2 QCS-NMR positivity (≤0.56) was associated with improved PFS with Dato-DXd compared to docetaxel (6.9 months vs. 4.1 months, *p* = 0.0063) [198].

TROPION-Lung02 is currently evaluating the combination of Dato-DXd with pembrolizumab with or without platinum-based chemotherapy. It includes both previously treated metastatic NSCLC with <2 lines of therapy and treatment-naïve patients [199]. Consistent with phase I studies, the most common AEs included nausea, stomatitis and neutropenia, increased amylase, and drug-induced interstitial lung disease [199]. The study is still ongoing, and further results are anticipated to provide more comprehensive data on the efficacy and safety of Dato-DXd in this patient population. There are several similar ongoing clinical trials assessing a variety of combination therapies with Dato-DXd. TROPION-Lung04 is a phase 1b study including treatment-naïve advanced NSCLC patients without actionable genomic alterations exposed to Dato-DXd with immunotherapy ± carboplatin [NCT04612751]. The trial included multiple cohorts, each with a different immunotherapy or PD-L1 TPS score, including durvalumab, AZD2936 (anti-PD-1/TIGIT), and MEDI5752 (anti-PD-1/CTLA-4). Alternatively, TROPION-Lung05 is a phase 2 study evaluating Dato-DX in patients with actional genomic alterations who progressed on prior targeted or platinum chemotherapy, which is showing promising results, with an ORR of 35.8% [NCT04484142]. Other ongoing clinical trials include a variety of combinations of Dato-DXd with osimertinib [NCT06417814, NCT06350097], rilvegostomig [NCT06357533], and pembrolizumab [NCT05555732, NCT05215340].

## 11. Conclusions

In conclusion, the treatment landscape for NSCLC is undergoing transformative advancements with the development of targeted therapies aimed at specific oncogenic drivers, as summarized in Figure 2. This review underscores the progress made in targeting mutations in KRAS, EGFR, HER2, MET, ALK, RET, ROS1, TROP-2, and FGFR, which enhances treatment options and outcomes for patients with traditionally limited therapeutic alternatives. The efficacy demonstrated by next-generation inhibitors and ADCs represents a major shift toward personalized medicine. It also highlights the importance of molecular testing at the time of initial diagnosis and at the time of progression to better characterize targetable driver mutations or acquired resistance mechanisms. Continued research into combination strategies, resistance mechanisms, and biomarker-driven therapies will be essential to further refine these targeted treatments. The future of NSCLC therapy lies in precision oncology, intending to tailor treatments to each patient’s unique tumor profile for improved survival and quality of life.

## Figures and Tables

**Figure 1 cancers-17-00353-f001:**
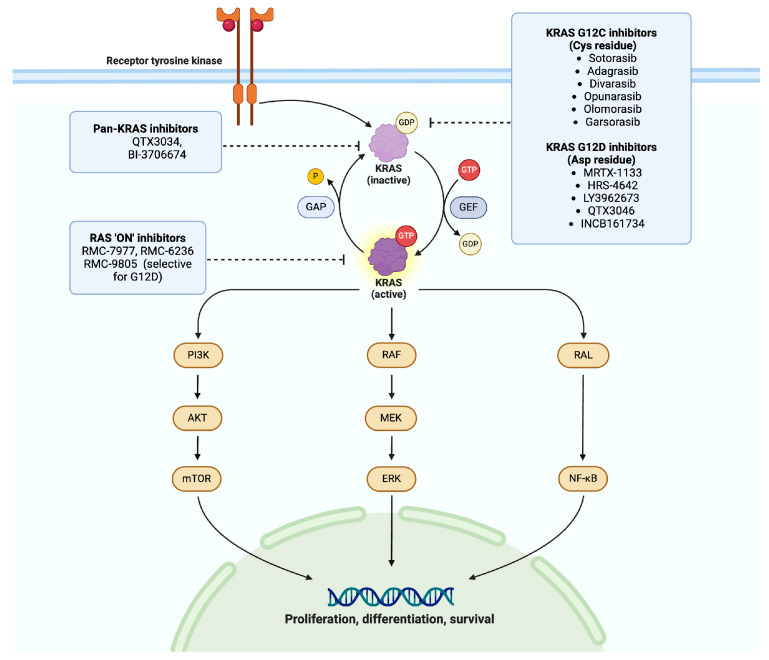
Summary of emerging KRAS inhibitors in NSCLC. Created using biorender.com.

**Figure 2 cancers-17-00353-f002:**
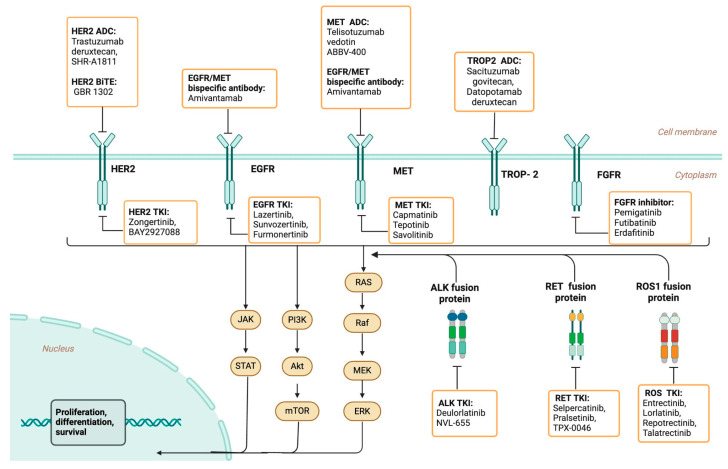
Summary of emerging targeted therapies in NSCLC. Created using biorender.com.

**Table 1 cancers-17-00353-t001:** Summary of genetic alterations in NSCLC and respective targeted therapies.

Genetic Alteration	Adenocarcinoma	Squamous	Other	Targeted Therapeutics
**KRAS**				
G12C mutation	13–17%	2–4%	Rare	sotorasib, adagrasib divarasib, opunarasib olomorasib, garsorasib
G12D mutation	14–18%	1–1.5%	Rare	MRTX-1133, HRS-4642, LY3962673 RMC-9805, ASP3082
**EGFR**				
Ex19del	21%	3–5%	Rare	EGFR TKIs (1st–3rd generation)
Exon 21 L858R	29%	1–3%	Rare	EGFR TKIs (1st–3rd generation)
Exon20ins	2–3%	<1%	Rare	amivantamab, furmonertinib sunvozertinib
Amplification	10–15%	15–20%	Rare	EGFR TKIs (1st–3rd generation)
Overexpression	40–60%	60–89%	Rare	Not predictive of response to TKI
**HER 2**				
Mutation exons 18–21	1–4%	<1%	Rare	HER2 TKIs (zongertinib, BAY2927088), ADC (TDXd SHR-A1811), monoclonocal antibody and BiTEs
Amplification	2–5%	1–2%	Rare	ADC (TDXd SHR-A1811), monoclonocal antibody and BiTEs
Overexpression	6–30%	2–10%	Rare	ADC (TDXd SHR-A1811), monoclonocal antibody and BiTEs
**MET**				
METex14 skipping mutations	3–4%	1–2%	15–20% in sarcomatoid	TKIs crizotinib, cabozantinib, capmatinib, tepotinib) and MET-ADC (Teliso-V, REGN5093-M114)
Amplification	1–5% (High-level <1%)	2–5%	~1–2%	TKIs crizotinib, cabozantinib, capmatinib, tepotinib) and MET-ADC (Teliso-V, REGN5093-M114)
Overexpression	15–50%	20–60%	Rare	MET-ADC (Teliso-V, REGN5093-M114)
**ALK**				
Fusion	3–7%	~0.1–0.3%		ALK inhibitors (e.g., crizotinib, alectinib, brigatinib, lorlatinib).
Amplification	~1–3%	~0.5–1%		Less predictive of response
**ROS**				
Fusion	1–2%	<0.1%	Rare	Multitargeted TKIs (e.g., lorlatinib, entrectinib, repotrectinib, taletrectinib)
Amplification	~1–2%	<1%	~1–2%	
**RET**				
Fusion	1–2%	<0.1%	Rare	RET-targeting inhibitors like selpercatinib and pralsetinib.
**FGFR**				
Fusion	1–2%	<0.1%		FGFR inhibitors, such as erdafitinib, pemigatinib, and infigratinib.
**TROP2**				
Overexpression	20–30%	10–20%	Rare	Therapeutic antibodies (e.g., sacituzumab govitecan

**Table 2 cancers-17-00353-t002:** List of ongoing clinical trials involving KRAS inhibitors.

NCT Number	Brief Description of Current Trials with KRAS Inhibitors
NCT05374538	VIC-1911 monotherapy with sotorasib for treatment of KRAS G12C-NSCLC
NCT05074810	Phase I/II Study of avutometinib (VS-6766) and sotorasib with or without Defactinib in KRAS G12C NSCLC
NCT05840510	Adagrasib with nab-Sirolimus in advanced solid tumors and NSCLC with KRAS G12C mutation (KRYSTAL -19)
NCT05578092	Phase 1/2 Study of MRTX0902 (SOS1 inhibior) in solid tumors mutations in the KRAS MAPK Pathway
NCT05737706	Study of MRTX1133 (G12D inhibitor) in advanced solid tumors with KRAS G12D mutation
NCT04975256	Phase I/Ib trial of adagrasib with BI 1701963 in advanced solid tumors with KRAS G12C mutation
NCT05358249	Platform Study of JDQ443 (G12C inhibitor) in select combinations of with either trametinib, ribociclib or cetuximab in advanced solid tumors with KRAS G12C mutation
NCT06300177	D-1553 (Garsorasib) tablet vs docetaxel for KRAS G12C mutation in advanced or metastatic NSCLC
NCT05485974	Dose escalation study of HBI-2438 (G12C inhibitor) in solid tumors with KRAS G12C mutation
NCT05288205	Phase I/IIa study of JAB-21822 plus JAB-3312 in patients with advanced solid tumors with KRAS G12C mutation
NCT04699188	Study of JDQ443 (G12C inhibitor) in advanced solid tumors with KRAS G12C mutation
NCT06403735	Phase I trial of QLC1101 (G12D inhibitor) in advanced solid tumors
NCT05375994	Avutometinib (VS-6766) with adagrasib in KRAS G12C mutated NSCLC
NCT06130254	Phase Ib trial of adagrasib (MRTX849) in combination with olaparib (PARP inhibitor) in KRAS G12C mutated advanced solid tumors with a focus on gynecological, breast, pancreatic and KEAP1 mutated NSCLC
NCT06497556	Evaluating the efficacy and safety of divarasib vs sotorasib or adagrasib in previously treated KRAS G12C-positive advanced or metastatic NSCLC
NCT06026410	KO-2806 (farnesyl transferase inhibitor) monotherapy and combination therapies in advanced solid tumors
NCT05786924	Study of BDTX-4933 in KRAS, BRAF and Select RAS/MAPK mutated cancers
NCT05492045	Study including D-1553 (Garsorasib) in combination therapy in NSCLC
NCT03415126	Study of ASN007 (ERK 1/2 inhibitor) in advanced solid tumors
NCT01951690	Phase II study of VS-6063 (defactinib) in KRAS mutant NSCLC
NCT05375084	SHP2 Inhibitor BBP-398 in combination with nivolumab in advanced NSCLC with KRAS mutation
NCT01362296	Open-label Study of GSK1120212 (trametinib) compared with docetaxel in stage IV KRAS-mutant NSCLC
NCT04916236	Combination therapy of RMC-4630 (SHP2 inhibitor) and LY3214996 (ERK inhibitor) in metastatic KRAS mutant solid tumors
NCT05480865	SHP2 Inhibitor BBP-398 in combination with sotorasib in advanced solid tumors with KRAS-G12C mutation
NCT06078800	Study of YL-17231 (oral pan-RAS inhibitor) in advanced solid tumors
NCT06162221	Study of RAS(ON) Inhibitor combinations in advanced RAS-mutated NSCLC
NCT05631574	Study of covalent menin inhibitor BMF-219 in KRAS mutant NSCLC, pancreatic, and colorectal cancer
NCT06585488	First-in-human study of BGB-53038 (Pan-KRAS Inhibitor) Alone or in combinations in advanced or metastatic solid tumors with KRAS mutations or amplification
NCT03990077	Study of HL-085 (MEK inhibitor) plus docetaxel in KRAS Mutant NSCLC

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
