# Peer review of "Emerging Targeted Therapies in Non-Small-Cell Lung Cancer (NSCLC)"

_cancers, 2025, doi:10.3390/cancers17030353_

Round 1
Reviewer 1 Report
Comments and Suggestions for Authors
This review provides a comprehensive summary of emerging targeted therapies for commonly encountered actionable mutations in NSCLC, with a focus on KRAS, EGFR, ALK, MET, HER2, RET, FGFR, and TROP2. It also reviews recent clinical trials, detailing their designs, patient responses, and associated adverse effects. Overall, the manuscript is well-structured, informative, and supported by appropriate figures and charts that enhance clarity and understanding. I suggest the following minor revisions before the manuscript proceeds to publication:
1. While the manuscript focuses on KRAS, EGFR, ALK, MET, HER2, RET, FGFR, and TROP2, which are undeniably important, other critical therapeutic targets in NSCLC, such as RAF, MEK, and ROS1, are not included in detail. It would be helpful to explain in the introduction why these targets were not addressed, as this rationale will clarify the scope and focus of the review without diminishing its impact.
2. For clarity and consistency, renumber the KRAS section as 1. KRAS (line 61)
3. Line 68-70: In citation 12, the authors analyzed the incidence of KRAS exon 2 mutant alleles in LUAD, reporting a prevalence approaching 40% for KRAS G12C. Please revise the description to align accurately with the data presented in citation 12.
4. Line 136: Correct the typo in "Q61X."
5. Line 185: Missing a space following QTX3034.
6. Line 497-501: Provide citations for Defloration and the associated clinical trials.
7. Line 574-677: Several statements lack citations. Please carefully review the manuscript to ensure that all references to previous research are appropriately cited and formatted.
8. Figure 2: Include information about FGFR in the figure to align with the comprehensive discussion in the text.
Author Response
|
Comments 1: While the manuscript focuses on KRAS, EGFR, ALK, MET, HER2, RET, FGFR, and TROP2, which are undeniably important, other critical therapeutic targets in NSCLC, such as RAF, MEK, and ROS1, are not included in detail. It would be helpful to explain in the introduction why these targets were not addressed, as this rationale will clarify the scope and focus of the review without diminishing its impact.
|
|
Response 1: Thank you for pointing this out. We agree with this comment, in particular the importance of including ROS1 mutated NSCLC in the discussion. However, given there are no significant breakthrough emerging drugs for the RAF/MEK pathway, those were considered outside the scope of this review. The following revisions have been made: - In the introductory paragraph, addition of a sentence explaining why RAF/MEK are out of scope. Page 2, line 61-63 - We have added a new paragraph to highlight the most recent breakthroughs for ROS1. Revisions can be found as the paragraph numbered 6, page number 13, line number 594.
|
|
Comments 2: For clarity and consistency, renumber the KRAS section as 1. KRAS (line 61) |
|
Response 2: This has been revised in the manuscript. |
|
Comments 3: Line 68-70: In citation 12, the authors analyzed the incidence of KRAS exon 2 mutant alleles in LUAD, reporting a prevalence approaching 40% for KRAS G12C. Please revise the description to align accurately with the data presented in citation 12. |
|
Response 3: This has been revised in the manuscript in line 72, red font. |
|
Comments 4: Line 136: Correct the typo in "Q61X |
|
Response 4: This has been revised in the manuscript in page 4, line 142 |
|
Comments 5: . Line 185: Missing a space following QTX3034. |
|
Response 5: This has been revised in the manuscript in page 5, line 191 |
|
Comments 6: Line 497-501: Provide citations for Defloration and the associated clinical trials. |
|
Response 6: Assuming this was a typo where ‘Defloration’ was originally meant to be ‘Deulorlatinib’, appropriate citation for the phase I study has been added, revision is in red. Page 12 Line 587. |
|
Comments 7: Line 574-677: Several statements lack citations. Please carefully review the manuscript to ensure that all references to previous research are appropriately cited and formatted. |
|
Response 7: Additional citations have been added for the respective paragraphs. Revisions are marked in red. Lines 707-711 |
|
Comments 8: Figure 2: Include information about FGFR in the figure to align with the comprehensive discussion in the text.
|
|
Response 8: Figure 2 has been edited to add FGFR and ROS fusion protein as well. Uploaded as supplementary file and included in the manuscript as well. |
Reviewer 2 Report
Comments and Suggestions for Authors
This is a well written, clinically relevant review of targeted treatments for patients with metastatic NSCLC and actionable genomic alterations, which is up to date. The authors exhibited good judgement whilst including the most relevant information, they have not made this too long and difficult for the reader to read.
Essentially this is a narrative review, and it may be appropriate to comment, as to the literature review process followed, e.g. covering molecular biology aspects / established treatments / approved drugs and recent conferences eg WLCC 2024 for new agents in phase I/II studies / drug development.
May be appropriate for EGFR beyond exon 20 insertions, to have at least 2-3 lines or a small paragraph for the other uncommon EGFR mutations.
Reading through the document, for some biomarkers, there is some information missing as to the incidence of the different genomic alterations eg for HER 2 or FGFR or MET etc. This would be nice to include in the manuscript, as it would not take a lot of space to do this, plus it would be nice for the reader to have the relevant information available, and to highlight the impact of the different genomic abnormalities for the same target in relation to the treatment.
Hence, it may be appropriate to consider a table where for each biomarker they quote the genomic abnormalities and % incidence, eg HER 2 mutations vs HER 2 amplification vs HER 2 overexpression, or MET exon 14 deletion vs amplification vs overexpression vs fusion, plus any differences in the incidence of the specific genomic abnormality according to histology eg MET exon 14 deletion with sarcomatoid histology. Plus in neighbouring columns having the approved drugs and drugs in development for each genomic abnormality (eg mutation, vs amplification vs high IHC) would be useful.
Author Response
|
Comments 1: Essentially this is a narrative review, and it may be appropriate to comment, as to the literature review process followed, e.g. covering molecular biology aspects / established treatments / approved drugs and recent conferences eg WLCC 2024 for new agents in phase I/II studies / drug development.
Response 1: Thank you for pointing this out. We agree with this comment. Therefore, we have added two sentences describing our literature review process in the third paragraph of the introduction. Revisions are in red, page 2, line 53-56
Comments 2: May be appropriate for EGFR beyond exon 20 insertions, to have at least 2-3 lines or a small paragraph for the other uncommon EGFR mutations.
Response 2: Thank you for pointing this out. We agree with this comment. Therefore, we added a full paragraph at the end of EGFR summary to discuss EGFR uncommon mutations. Revisions are in red, page number 8, line 388-405
Comments 3: Reading through the document, for some biomarkers, there is some information missing as to the incidence of the different genomic alterations eg for HER 2 or FGFR or MET etc. This would be nice to include in the manuscript, as it would not take a lot of space to do this, plus it would be nice for the reader to have the relevant information available, and to highlight the impact of the different genomic abnormalities for the same target in relation to the treatment. Hence, it may be appropriate to consider a table where for each biomarker they quote the genomic abnormalities and % incidence, eg HER 2 mutations vs HER 2-amplification vs HER 2 overexpression, or MET exon 14 deletion vs amplification vs overexpression vs fusion, plus any differences in the incidence of the specific genomic abnormality according to histology eg MET exon 14 deletion with sarcomatoid histology. Plus, in neighbouring columns having the approved drugs and drugs in development for each genomic abnormality (eg mutation, vs amplification vs high IHC) would be useful.
Response 3: Thank you for pointing this out. We agree with this comment and therefore a summary table 1 has been added to the manuscript. Including every genetic alteration type for each gene resulted into a lengthy table, therefore the genetic alterations included are limited to the ones more pertinent in this review article.
|
Reviewer 3 Report
Comments and Suggestions for Authors
Dr. Parikh and the team have authored a timely and insightful review article examining the role of oncogenic mutations in lung cancer, with a focus on biological mechanisms, clinical implications, and future therapeutic strategies. While the review is well-written and holds significant translational value, a few points need to be addressed to enhance its comprehensiveness:
-
Several studies have shown that the YAP/TEAD pathway is a key contributor to resistance against KRAS G12C and KRAS G12C + SHP2 inhibitor combinations in lung cancer. Whole-genome CRISPR screens have further validated additional critical pathways, such as PI3K and mTOR, alongside YAP/TEAD. The authors should include a discussion on this topic and reference these important studies (PMID: 37729426 and PMID: 37934115) to keep the review current and well-rounded.
-
Cancer metabolism plays a definitive role in oncogenic signaling pathways, as highlighted in recent literature (PMID: 33870211 and PMID: 38841622). Targeting cancer metabolism is an essential approach for improving the efficacy of small molecule therapies in various cancers, including lung cancer. The authors should address this aspect, citing relevant studies to provide a more comprehensive perspective.
- Recent reports indicate that clinical trials involving SHP2 inhibitors from several companies have been halted or terminated due to toxicity or other issues when used in combination therapies for lung cancer. Notable examples include trials from Sanofi/RevMed, Pfizer/Array, BMS/BridgeBio, Genentech/Relay, Novartis, and Erasca. The authors should include a brief discussion on this current development, offering their perspective on SHP2 inhibitor-based therapeutic strategies in lung cancer, as this insight could be valuable for planning future treatment approaches.
Author Response
|
Comments 1: Several studies have shown that the YAP/TEAD pathway is a key contributor to resistance against KRAS G12C and KRAS G12C + SHP2 inhibitor combinations in lung cancer. Whole-genome CRISPR screens have further validated additional critical pathways, such as PI3K and mTOR, alongside YAP/TEAD. The authors should include a discussion on this topic and reference these important studies (PMID: 37729426 and PMID: 37934115) to keep the review current and well-rounded.
|
|
Response 1: Thank you for pointing this out. These are certainly very important studies, and we have included both 37729426 and 37934115 in the discussion for resistance mechanisms for KRAS inhibitors. Revisions can be found in red font on page 3, lines 133-136 Comments 2: Cancer metabolism plays a definitive role in oncogenic signaling pathways, as highlighted in recent literature (PMID: 33870211 and PMID: 38841622). Targeting cancer metabolism is an essential approach for improving the efficacy of small molecule therapies in various cancers, including lung cancer. The authors should address this aspect, citing relevant studies to provide a more comprehensive perspective. Response 2: Thank you for pointing this out. We certainly appreciate the importance of these studies in precision oncology. However, since the focus of this review is to highlight the emerging therapeutic options applicable in clinical practice, we felt these would be out of scope for the practicing clinicians. Comments 3: Recent reports indicate that clinical trials involving SHP2 inhibitors from several companies have been halted or terminated due to toxicity or other issues when used in combination therapies for lung cancer. Notable examples include trials from Sanofi/RevMed, Pfizer/Array, BMS/BridgeBio, Genentech/Relay, Novartis, and Erasca. The authors should include a brief discussion on this current development, offering their perspective on SHP2 inhibitor-based therapeutic strategies in lung cancer, as this insight could be valuable for planning future treatment approaches.
Response 3: Thank you for pointing this out. We agree with this comment. Therefore, we have added an additional paragraph discussing the role of SHP2 inhibitors and some of the current updates/challenges encountered with them. Revisions are in red font on page 5, lines 221-238. |
Reviewer 4 Report
Comments and Suggestions for Authors
The article provides a comprehensive overview of treatments for lung cancer, including existing strategies and therapeutic options. The manuscript is well-organized, and the information is presented clearly, making it accessible to readers. Good points:
- The authors have included a broad spectrum of existing treatments, offering a solid foundational understanding for readers new to the field.
- The manuscript demonstrates good synthesis of well-established knowledge, which can be useful for educational purposes or as a quick reference.
Problems:
While the manuscript achieves its goal of summarizing treatments, it falls short in contributing novel insights or perspectives. The field of lung cancer treatment has been extensively reviewed in recent years, with numerous publications addressing the same type of information presented in this article As such, this article does not significantly add to the existing body of knowledge.
For future revisions or follow-up work, I would recommend the authors focus on:
- Offering critical evaluations or meta-analyses of current therapies to provide unique insights or consolidate data for a broader impact.
Conclusion:
The article serves as a useful general review but lacks novelty in the context of recent literature.
The article does not provide novelty in terms of review type. There are already plenty of reviews published on the same subject.
Author Response
|
Comments 1: While the manuscript achieves its goal of summarizing treatments, it falls short in contributing novel insights or perspectives. The field of lung cancer treatment has been extensively reviewed in recent years, with numerous publications addressing the same type of information presented in this article. As such, this article does not significantly add to the existing body of knowledge. For future revisions or follow-up work, I would recommend the authors focus on: Offering critical evaluations or meta-analyses of current therapies to provide unique insights or consolidate data for a broader impact.
|
|
Response 1: Thank you for pointing this out. We certainly appreciate the constructive feedback. Considering the comments from other reviewers as well, we have added several new discussion points to further enhance the quality of the paper. To add clinical context to the paper, we attempted to share our interpretation of the data and practice preferences to aid in decision-making for the practicing clinician. These perspectives are included at the end of each section. Revision were made to the following sections (KRAS< EGFR, HER2, MET, ALK, RET, and TROP2). Revisions are in red font. Lines: 248-257 329-345 462-464 531-533 570-577 698-704 791-798
|
Round 2
Reviewer 3 Report
Comments and Suggestions for Authors
Accepted
Reviewer 4 Report
Comments and Suggestions for Authors
I think the revised version is acceptable now.